

# A new species of *Languidipes* Hubbard (Ephemeroptera, Polymitarcyidae) from Borneo

Guillermo Eduardo Hankel[1,2] and Carlos Molineri[1]

[1] Instituto de Biodiversidad Neotropical (IBN), Consejo Nacional de Investigaciones Científicas y Técnicas (CON-ICET), Facultad de Ciencias Naturales e Instituto Miguel Lillo, Universidad Nacional de Tucumán, Yerba Buena, Tucumán, Argentina

[2] Instituto de Invertebrados, Fundación Miguel Lillo, San Miguel de Tucumán, Tucumán, Argentina

## ABSTRACT

The genus *Languidipes* is currently represented by three species distributed in south-eastern Asia, India, and Sri Lanka. *Languidipes corporaali* is the most widely distributed species, and both, male and female imagos, as well as nymphs, are known. In contrast, the other species, *L. taprobanes* and *L. lithophagus*, are only known from nymphs. Here, we describe a new species, *Languidipes janae* sp nov, based on male imagos collected from Borneo, Indonesia. This new species is characterized by the presence of ommation on mesonotum, and penis almost completely divided, with sub-quadrate base and a small outer projection basal to the long and slender distal arms. This constitutes the first record of the genus for Borneo. A cladistic analysis of the subfamily Asthenopodinae supports its taxonomic status.

## INTRODUCTION

Polymitarcyidae (Ephemeroptera), with a worldwide distribution, includes large to medium-sized mayflies with burrowing nymphs (*Kluge, 2004*; *McCafferty, 2004*). The mandibular tusks of the immature forms are used to dig tunnels in a variety of underwater sediments, including mud, clay and even siliceous rocks (*Molineri, Salles & Peters, 2015*; *Bolotov et al., 2022*). In addition they produce silk from the malpighian ducts, allowing them to coat their tunnels with a thin mesh of this material (*Sattler, 1967*), or even to construct silk cases where tunnels are impossible to dig (*Molineri & Emmerich, 2010*; *Pai et al., 2023*). Furthermore, adults are so short-lived that they do not present functional legs (except for the male forelegs, used to grasp females during copula), spending their entire life in flight. This forces them to make their subimaginal molt in a unique manner, not shading their cuticle in the classic form (as an entire piece) but in flakes that come off the body and wings (*Molineri, 2010*). Because of their unique biology, including nymphs hidden in the substrates and extremely short-lived adults, specimens of this group are infrequently collected.

Corresponding author
Guillermo Eduardo Hankel, guille-hankel@gmail.com

The genus *Languidipes* was originally described as *Asthenopus corporaali* (*Lestage, 1922*) from Java, Indonesia. *Languidipes corporaali* (Lestage) was subsequently recorded from other Indonesian localities (Sumatra and Simeulue), as well as from Malaysia and Thailand (*Baumgardner et al., 2012*). The genus *Languidipes* also includes the species *L. taprobanes* (*Hubbard, 1984*) (*Hubbard, 1984*; *Rathinakumar, Kubendran & Balasubramanian, 2019*; *Pai et al., 2023*), from India and Sri Lanka, and the recently described *L. lithophagus* (*Bolotov et al., 2022*) from Myanmar.

A phylogenetic framework has been proposed for the subfamily Asthenopodinae, where *Languidipes* is included together with partially sympatric *Povilla* and other three South American genera (*Molineri, Salles & Peters, 2015*).

Here we describe a new species of *Languidipes* based on male imagos from Borneo, Indonesia, and test its phylogenetic relationships inside the subfamily.

## MATERIALS & METHODS

Specimens were fixed in 70° % (v/v) ethanol. One wing was removed and mounted dry on microscope slides. Genitalia was dissected and temporarily mounted in gel alcohol for study and drawings with a camera lucida attached to an Olympus BX51 microscope. Photographs were taken with a Zeiss Axiocam ICc5 attached to a Zeiss Stemi 508 stereo microscope. Some images were processed with CombineZP software (*Hadley, 2010*) to improve focus.

Material is deposited in the following Institution: IBN (Instituto de Biodiversidad Neotropical, Tucumán), and FAMU (Florida A&M University, Tallahassee, FL).

The morphological matrix published in *Molineri, Salles & Peters (2015)* was revised, the new species amended, and some characters of *L. corporaali* were modified following the description of *Baumgardner et al. (2012)*. All other taxa and characters in the matrix were not modified (Appendix 1).

The TNT program (*Goloboff, Farris & Nixon, 2008*) was used to set up the most parsimonious trees. Heuristic searches were conducted under implied weights (*Goloboff, Mattoni & Quinteros, 2006*) with $k = 3$ and 100 replicates of tree bisection and reconnection. All characters were treated as non-additive except for continuous characters (chars. 0 to 26), for additional details see *Molineri, Salles & Peters (2015)*. Group support was calculated with the method of frequency difference (*Goloboff et al., 2003*), using 1,000 replications of symmetric jackknifing.

The electronic version of this article in Portable Document Format (PDF) will constitute a published work as defined by the International Commission on Zoological Nomenclature (ICZN). Consequently, the new names introduced in the electronic version are deemed effectively published under the Code solely from the electronic edition. This published work, along with the associated nomenclatural acts, has been registered in ZooBank, the online registration system for the ICZN. The ZooBank LSIDs (Life Science Identifiers) can be accessed and the relevant information viewed through any standard web browser by appending the LSID to the prefix http://zoobank.org/. The LSID for this publication is: [LSIDurn:lsid:zoobank.org:act:048403BC-2E75-4C1B-AE70-8DDF826FF9CA]. The

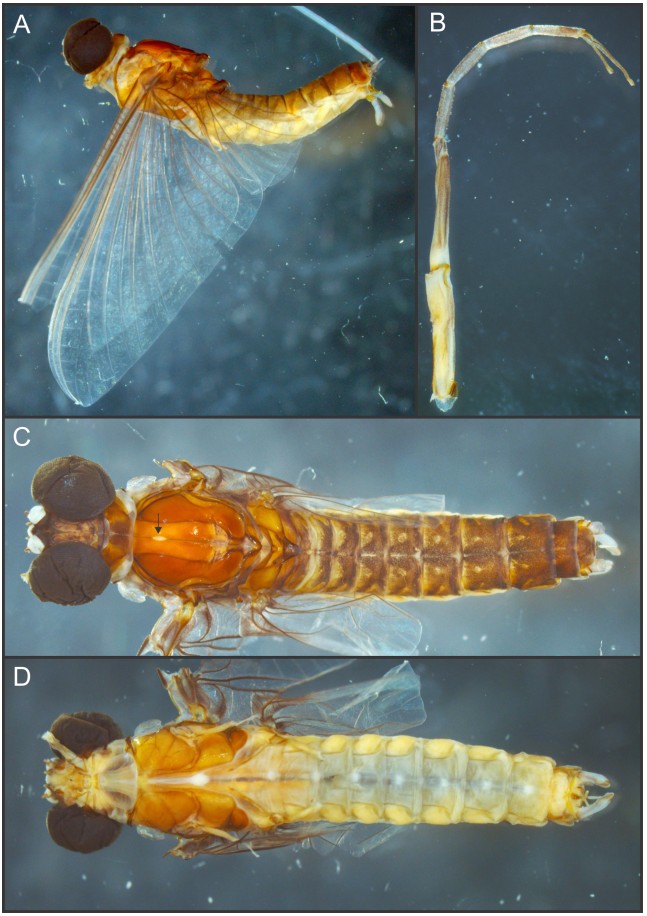

**Figure 1** *Languidipes janae* **sp. nov., male imago.** (A) Lateral habitus; (B) foreleg, dorsal; (C) dorsal habitus (wings removed); (D) ventral habitus (wings removed). Photo by Carlos Molineri.

online version of this work is archived and available from the following digital repositories: PeerJ, PubMed Central SCIE, and CLOCKSS.

# RESULTS

Description
*Languidipes janae* sp. nov. (Figures 1 –3)

Type material. Holotype male imago from Indonesia (Borneo): Kalimantan, Timur Prov., Lake Semayang, nr. Kota Bangun, attracted to light on boat, 3.vii.1985, M. Christensen, specimen number IBN–E 6370. Paratypes: four male imagos, same data, all deposited in IBN (IBN–E–6371, IBN–E–6372, IBN–E–6373 and IBN–E–6374).

Additional material. We also examined 1 larvae of *L. taprobanes*, paratype, FAMU E2109, from Ceylon, Kollonawe, iv.1954 (no more data).

Diagnosis. The male imago of this species is characterized by the presence of ommation on mesonotum, and penis divided almost completely, with sub-quadrate base, small outer projection basally to the long and slender distal arms; distal arms with pointed apex.

Male imago. Length (mm): body, 10.0–14.0; forewing, 12.2–13.0; hind wing, 4.0–5.0; cercus, 26.0, parecercus, 0.5−1.1. Head. Compound eyes large, black, covering most of head, separated in the middle of head by a distance equal to 1/3 of the width of an eye (Figs. 1A, 1C); lateral ocelli large and pedunculated (Fig. 1C). Head brown dorsally, shaded with black mainly at the base of ocelli; ventrally much paler. Remnants of mouthparts whitish yellow. Antenna: scape and pedicel yellowish (flagellum broken-off and lost). Thorax. Pronotum reddish brown with black stippling on central area; anterior membranous portion blackish, posterior margin withish; sternum and pleura whitish. Mesonotum reddish brown slightly paler medially, shaded with black between the posterior scutal protuberances; ommation (oval whitish median area in anterior $\frac{1}{4}$ of mesonotum) present (arrow in Fig. 1C); pleura and sternum light yellowish brown, furcasternal median impression translucent. Metanotum reddish brown shaded with black on median area and posterior margin, pleura yellowish, sternum whitish translucent. Forelegs relatively short (slightly shorter than $\frac{1}{2}$ of body length), yellowish white (Fig. 1B). Middle and hind legs whitish, weak (Fig. 1D). Forewings (Fig. 2A) hyaline shaded with gray along costal margin and on membrane basal to vein A. Hindwings (Fig. 2A) hyaline, shaded with gray at costal and basal half of subcostal areas, and at base. Veins of both wings brownish, lighter toward apex, except cross veins on apical half of wing, translucent. Abdomen. Dorsum brownish shaded with black, ventrally whitish. Genitalia (Figs. 2B to 2E, 3A and 3B): forceps one-segmented, robust, distally with a patch of short and curved setae along the inner margin. Penis divided almost completely, penis base sub-quadrate with a small outer projection (arrow in Figs. 2E and 3B), distal arms long and slender with pointed apex. Cerci: whitish, shaded with light gray basally. Paracercus as long as tergum X, whitish and thin.

Etymology. The specific name (noun in the genitive case) is a tribute to Janice Peters ("Jan"), who facilitated the material of the new species, and for her constant support.

Notes. In forewings, ICu veins presented variations among specimens. Frequently ICu1 is basally fused to CuA but may be basally free or joined to ICu2, additionally ICu2 may be basally free or fused to CuP.

Distribution. Data here presented constitute the first record of a *Languidipes* species in Borneo Island (Fig. 4).

Phylogenetic study

Only one shortest tree was recovered (Fig. 5), with a tree length of 270.8, a total fit of 5.8, and an adjusted homoplasy of 15.2. A high support was obtained for *Languidipes* (95%) and for the sister group *Languidipes* + *Povilla* (87%). The synapomorphies supporting the genus *Languidipes* (two species included) are: 1) ratio length second foretarsite/foretibia (char. 1 changes from 0.584−0.645 to 0.480), 2) ratio FW/foreleg length (char. 2, from 1.661−1.736 to 2.800), 3) ratio FW/cercus length (char. 3, from 0.339−0.347 to 0.375−0.464), 4) FW

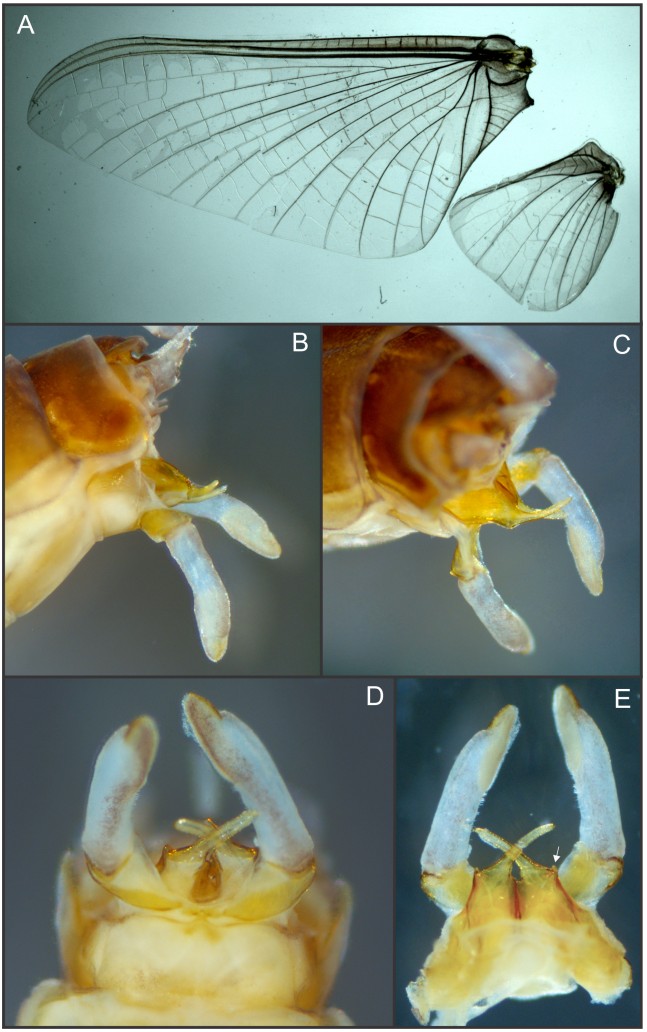

**Figure 2** *Languidipes janae* **sp. nov., male imago.** (A) Wings; (B) genitalia, lateral; (C) same, latero-dorsal; (D) ventral; (E) dorsal. Photo by Carlos Molineri.

ratio length/width (char. 4, from 2.000−2.214 to 2.265), 5) ratio length FW/HW (char. 5, from 2.302−2.447 to 2.790), 6) penes, ratio basal width/subapical width (char. 17, from 1.300 to 2.000), 7) FW Cu sector, ICus joinning hind margin on different sides of tornus (char. 35): ICu1 close to tornus, ICu2 on basitornal margin, and 8) median plate of styliger (char 41) absent. The autapomorphies found for *Languidipes janae* are: 1) ratio subapical width of foretibia/subbasal width of tarsite 2 (char. 0, from 1.700 to 1.040), 2) ratio FW/cercus length (char. 3, from 0.375−0.464 to 0.500), 3) ratio marginal length between main longitudinal veins/imv length (mean of all values in a wing) (char. 9, from 1.653 to 1.745), 4) Rs stem length (FW male)/Rs from fork to margin (char. 10, from 0.235−0.241 to 0.220), 5) ratio total length of forceps/basal width (char. 13, from 4.545 to 4.300−4.500), 6) ratio length/basal width of penile lobe (char. 15, from 4.706−5.200 to
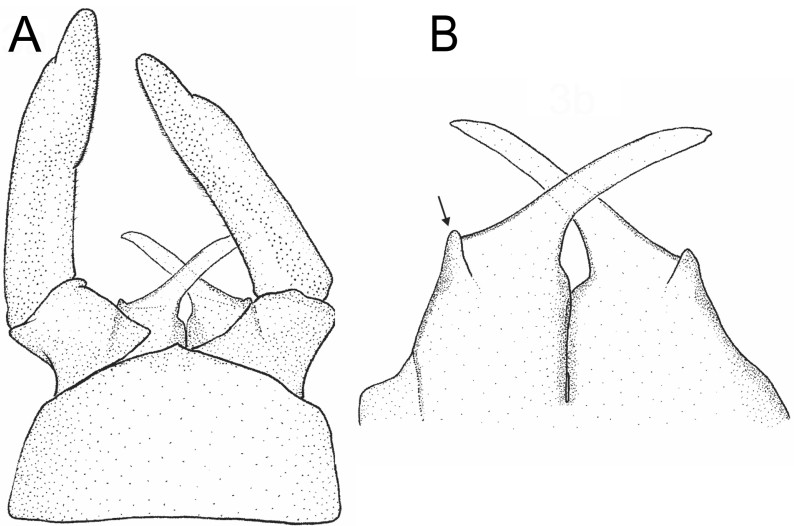

**Figure 3** *Languidipes janae* **sp. nov., male imago.** (A) Genitalia, ventral; (B) penis, dorsal. Illustration by Carlos Molineri.

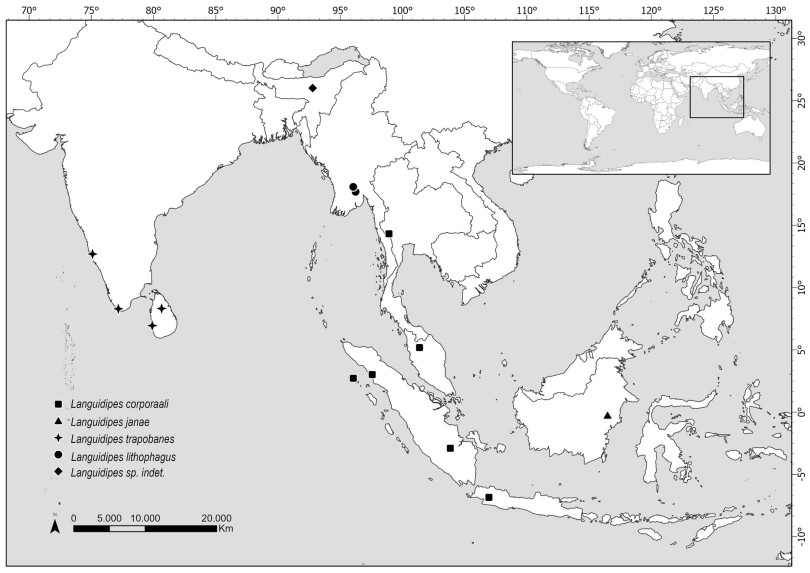

**Figure 4** **Map showing the distribution of all known *Languidipes* species.** Map elaborated by Luciana Cristobal. Map elaborated with QGIS 3.34. Made with Natural Earth, Free vector and raster map data (https://www.naturalearthdata.com).

2.600), 7) penes, ratio basal width/subapical width (char. 17, from 2.000 to 3.125), and 8) male foretarsite 1 subrectangular (char. 29).
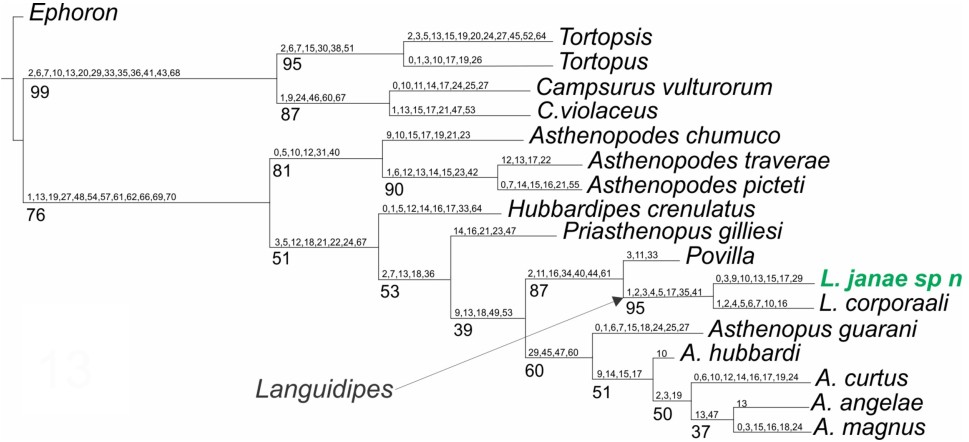

**Figure 5** Phylogenetic tree of the Asthenopodinae subfamily, incorporating *Languidipes janae*.

## DISCUSSION

The species of *Languidipes* seem restricted to southeastern Asia (Fig. 4). The range of *Languidipes corporaali* is the widest of the genus, being recorded in some Indonesian islands (Java, Sumatra, and Simeulue), Thailand, and Malaysia; with a doubtful record for Assam, India (*Chopra, 1927* cited in *Hubbard, 1984*). *Hubbard (1984)* affirms that probably this last record will be a new species.

Most species of *Languidipes* are only known from nymphs. *Languidipes taprobanes* is known from Sri Lanka and the south of India, while *L. lithophagus* was recently described from Myanmar (*Bolotov et al., 2022*). It is possible that the males described here as *L. janae* represent the adult stage of one of them, but this seems unlikely. Nevertheless, we prefer to describe the new species because it constitutes the unique record from Borneo, and its size is relatively smaller than the other species (*Hubbard, 1984*; *Rathinakumar, Kubendran & Balasubramanian, 2019*; *Bolotov et al., 2022*; *Pai et al., 2023*). While we believe that obtaining a DNA barcode would have been very helpful for the future association of nymphs or female adults, this technique couldn't be pursued due to the nature of the available material. The material is scarce and has been collected over a long period (more than 30 years), preserved in a manner that does not support genetic material preservation. Therefore, the utilization of this technique was ruled out.

Styliger in *Languidipes* is reduced to pedestals, which appear to be the basal segment of forceps. Median plate of styliger is not present, contrary to *Povilla* and other Asthenopodinae, but similar to Campsurinae (*Kluge, 2004*; *Molineri, Salles & Peters, 2015*). Following this interpretation, forceps of *Languidipes* are one-segmented, and the diagnosis proposed by *Baumgardner et al. (2012)* including the statement "male genitalia without a remnant of styliger plate" should be amended to "male genitalia without a remnant of the median plate of styliger".

Surprisingly, a weak small circular area in the center of the mesonotum (Fig. 1C) is present in the specimens here studied. This structure, much resembling the ommation of

Caenidae and Neoephemeridae (*Wang, McCafferty & Bae, 1997*), is unique in the family Polymitarcyidae, and most probably is an independent acquisition.

Among the species of *Languidipes*, only *L. corporaali* is known from the male adult, and it presents a penis structure strongly different to *L. janae* sp. nov. The basal portion of the penis are wide and laterodistally rounded in *L. corporaali* but is sub-quadrate and with an acute projection in outer margin in *L. janae*. Penis arms in *L. corporaali* end more acutely than in the species described here. Finally, the penis is divided from the base of the arms to the apex in *L. corporaali*, but *L. janae* presents a much deeper division including most of the basal portion of the penis.

The previous phylogenetic hypothesis (*Molineri, Salles & Peters, 2015*) is not modified by the inclusion of *Languidipes janae*. As expected, this species is grouped with *L. corporaali* in a well-defined group, sister to *Povilla*.

## ACKNOWLEDGEMENTS

We thank Luciana Cristobal for the map, and Janice Peters for providing the specimens here described.

### Funding
The authors received no funding for this work.

### Competing Interests
The authors declare there are no competing interests.

### Author Contributions
- Guillermo Eduardo Hankel conceived and designed the experiments, performed the experiments, analyzed the data, authored or reviewed drafts of the article, and approved the final draft.
- Carlos Molineri conceived and designed the experiments, performed the experiments, analyzed the data, prepared figures and/or tables, authored or reviewed drafts of the article, and approved the final draft.

### Data Availability
The character matrix used for the phylogenetic analysis is available in the Supplemental File.

### New Species Registration
The following information was supplied regarding the registration of a newly described species:
Publication LSID: urn:lsid:zoobank.org:pub:D4CB4BD7-422E-4628-8DFD-0BD18D5DC6B1

Genus name, Languidipes LSID: urn:lsid:zoobank.org:act:9EE2FF15-F1B4-47C7-AF2B-4AECBA0FAF1F

Species name, Languidipes janae
LSID: urn:lsid:zoobank.org:act:048403BC-2E75-4C1B-AE70-8DDF826FF9CA

## Supplemental Information

Supplemental information for this article can be found online at http://dx.doi.org/10.7717/peerj.17327#supplemental-information.

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
