# Peer review of "A new species of Languidipes Hubbard (Ephemeroptera, Polymitarcyidae) from Borneo"

_PeerJ, doi:10.7717/peerj.17327_

## Round 0.1 · original submission · Major Revisions

Please have a look at the reviewers and my comments on your paper. The major issue I see with your paper is the lack of DNA barcoding data. DNA barcoding nowadays is a major component of species identification and with the use of phylogenetic tree analysis normally provide strong evidence in support of species identification. This is especially true as you indicate in your paper your specimens might be the imagos of L. taprobanes or L. lithophagus. This leaves a big question mark without DNA barcoding data. If you can provide these data I am very willing to review the manuscript.

·

Basic reporting

This is a straight forward paper dealing with a new species of Languidipes from Borneo. The second author is the leading authority on this group.
The paper is well written, Pictures are of good quality. All needed references are cited.
The authors give sufficent arguments to prove the validity of their new species; the phylogenetic reconstruction confirms their hypothesis.
I think this paper casn be published with some minor corrections added in the draft.

Experimental design

no comment

Validity of the findings

This is the first mention of the genus Languidipes in Borneo; this finding is important to address biodiversity challenges in the future.

·

Basic reporting

I made only a few suggestions on text and figure plates over typos and taxonomic code.

Experimental design

I made a few suggestions and comments on description that I believe will help to improve it

Validity of the findings

no comment

Additional comments

no comment

---

## Round 0.2 · Minor Revisions

Thank you for responding to the reviews. I did have a few more minor edits for line 57, 58, and 69.

I understand your comments about DNA barcoding and you do provide strong evidence in the context of morphology, phylogeny, and geographic distribution.

Can you include a few sentences in your discussion acknowledging that DNA barcoding would have added additional support, however, due to the circumstances in points 1 and 2 you make, you are unable to include DNA barcoding.

---

## Round 0.3 · accepted · Accept

Thank you for making the additional changes and addressing previous comments by the reviewers. I believe the manuscript is ready for publication.